# Encapsulation of Sorghum Leaf Red Dye: Biological and Physicochemical Properties and Effect on Stability

**DOI:** 10.3390/foods12081646

**Published:** 2023-04-14

**Authors:** Carmo Serrano, Margarida Sapata, M. Conceição Oliveira, Andreia Soares, Carla Pereira, Rui M. V. Abreu, Lillian Barros

**Affiliations:** 1Instituto Nacional de Investigação Agrária e Veterinária (INIAV, I.P.), Av. da República, 2780-157 Oeiras, Portugal; 2LEAF|Linking Landscape, Environment, Agriculture and Food Research Unit, Associated Laboratory TERRA, Intituto Superior de Agronomia, Universidade de Lisboa, Tapada da Ajuda, 1349-017 Lisboa, Portugal; 3Centro de Química Estrutural, Institute of Molecular Sciences, Instituto Superior Técnico, Universidade de Lisboa, 1049-001 Lisboa, Portugal; 4Centro de Investigação de Montanha (CIMO), Instituto Politécnico de Bragança, Campus de Santa Apolónia, 5300-253 Bragança, Portugal; 5Laboratório Associado para a Sustentabilidade e Tecnologia em Regiões de Montanha (SusTEC), Instituto Politécnico de Bragança, Campus de Santa Apolónia, 5300-253 Bragança, Portugal

**Keywords:** encapsulation, 3-DXA, *Sorghum bicolor* L., biological properties, pH and thermal stability, spray-drying, freeze-drying

## Abstract

The encapsulation of the 3-deoxyanthocyanidins (3-DXA) red dye, extracted from sorghum (*Sorghum bicolor* L.) leaves, was explored for food application. The extracts showed antioxidant activity at concentrations ranging from 803 to 1210 μg mL^−1^ and did not reveal anti-inflammatory or cytotoxic properties, indicating their potential for food application. Encapsulation was performed with two carrier agents (maltodextrin and Arabic gum) in different proportions (1:1, 2:1 and 1.5:2.5 (*w*/*w*)). The microparticles produced by freeze-drying and spray-drying were studied according to the concentration of the dye, the encapsulation efficiency, the process yield, the solubility and the colour of the powders. The dye extracts are released from the microparticles at different pHs. The variation in ratio composition of the 3-DXA encapsulation was assessed by principal component analysis (PCA) using data from ten physicochemical parameters. The results indicated that the maltodextrin at the 2:1 ratio had a higher dye concentration and total phenolic content (TPC) at pH 6. This ratio was selected to produce the microparticles by freeze-drying and spray-drying, and the particles were used in the temperature stability tests at pH 6. The results suggest that the freeze-drying process offers better protection to 3-DXA, with a degradation percentage of 22% during the heating period (80 °C for 18 h), compared to the non-encapsulated dye (48%). However, there were no significant differences between the two polymeric agents. The non-encapsulated 3-DXA was evaluated as control and lost 48% of the total colour with the same treatment. Red dyes from sorghum leaf by-products may constitute promising ingredients for the food industry and increase the value of this crop.

## 1. Introduction

Sorghum (*Sorghum bicolor* (L.) Moench) is a crop native to East Africa, possibly Ethiopia, and is thought to have been domesticated around 1000 BC and adapted from the Atlantic coast to Ethiopia and Somalia. It is currently distributed between 50° N (USA and Russia) and 40° S, from sea level to an altitude of 1000 m [1].

During 2000–2015, the world sorghum production was 60 million tons. Africa, North America and Asia were the principal regions of production. Nigeria, the United States, Mexico, India and China are the top five largest producers, accounting for about 70% of world production. This is due to its versatility of cultivation in dry and hot areas with bad environmental conditions, where the productivity of other cereals is not economically feasible [2,3].

Currently, sorghum is considered the fifth most important cereal in terms of world production, preceded only by wheat, rice, maize and barley, due not only to the combination of genetic potential but also to the use of different cultivation practices, which have provided high grain and forage yields in regions and environmental conditions unfavourable for most cereals [4].

Some sorghum varieties are grown exclusively for grain (graniferous), while others are intended for forage production, and others have a dual purpose [5]. Grain sorghum varieties have different colours: red, orange, bronze, brown, white and black, and are used for food and feed as well as for the production of bioethanol. Forage sorghum varieties, on the other hand, are used for silage, sugar production, grazing/green cutting/wort roasting (biomass) and handicrafts. The vegetative part has high potential and can be used as hay or for the extraction of dye matter for various purposes, including dyeing of textiles and leather [6,7,8].

Sorghum phytochemicals, located mainly in the bran, forehead and pericarp layers of the grain, are simple phenols, phenolic acids (ferulic, p-coumaric, protocatechuic, vanillic, caffeic, p-hydroxybenzoic gallic and cinnamic acids), tannins, phytosterols, policosanols, carotenoids, lignans and flavonoids, including flavonoids, flavanones, flavones, isoflavones and anthocyanins that present human health benefits and can be used commercially [9,10].

Sorghum anthocyanins, termed 3-deoxyanthocyanis (3-DXA), comprise the luteolinidins and apigeninidins and their methoxylated derivatives, 5-methoxy-luteolinidin and 7-methoxy-apigeninidin [11,12,13] originating from plant secondary metabolites; they are often isolated by solvent extraction, highlighting their anticancer, anti-inflammatory, antimicrobial and antioxidant properties [9,12,13], and their potential as a dye in the food industry.

These anthocyanins have a low distribution in nature, being distinct from others due to the absence of an oxygen molecule in carbon 3 of this structure, which gives them stability to temperature variations compared to other anthocyanins found in fruits and vegetables [14]. However, this feature and its existence mainly as aglycones make these structures more hydrophobic and less stable in aqueous media [12,15].

The stability of 3-DXA in aqueous solutions increases with the addition of polymeric materials through the formation of complexes allowing extension of the application of these dyes in beverages, as verified by [15]. Other authors [16] studied the encapsulation of 3-DXA in esterified starch to offer food ingredients in powder form.

Anthocyanidins have been encapsulated in various carrier agents to increase stability under conditions of temperature, pH and light for food applications [17,18,19].

Several carbohydrate polymers have been used to encapsulate anthocyanins by spray-drying and freeze-drying in the food industry. Among them, maltodextrin is a mixture of saccharides with a wide molecular weight distribution between polysaccharides and oligosaccharides, and a dextrose equivalent (D.E.) of less than 20. Maltodextrins are products of the partial hydrolysis of starch from different botanical sources, obtained at low relative cost and neutral taste and aroma. They are characterised by high solubility in water, due to the presence of long carbohydrate chains, they exhibit a low viscosity at high concentrations, and have good film-forming ability and good protection against the oxidation of core materials. Arabic gum is non-toxic, odourless and tasteless, with excellent emulsification capacity, low viscosity in an aqueous solution and good volatile retention properties [20,21]. This behaviour is attributed to the highly branched and acidic polysaccharides of the molecular structure, so it is easily dispersed when stirred in water in concentrations up to 50% [22].

This work aimed to encapsulate the 3-DXA extracts, using two polymeric agents (maltodextrin and Arabic gum), by freeze-drying and spray-drying. The effect of the wall material on the encapsulation of 3-DXA was evaluated by releasing these compounds at different pHs and temperatures.

To the best of our knowledge, this is the first study using cell-based methods to assess the antioxidant activity of *S. bicolor*. Cytotoxicity was also evaluated to study the application of these dye powders as alternative natural dye options to replace certain azo dyes, such as Carmoisine (E122), Allura red (E129) or Ponceau 4 R (E124) in food and beverage reformulation in the food industry [23].

In this context, the sorghum leaves from agricultural valorisation production, currently an industrial surplus, were studied to transform them into natural colourants in powder form, at a low cost, that can be applied in the textile, cosmetic or food/feed industries.

## 2. Material and Methods

### 2.1. Chemicals and Materials

Ethanol p.a, hydrochloric acid (370 g·L^−1^), iron (II) sulphate heptahydrate, iron (III) chloride hexahydrate, sodium acetate trihydrate, were purchased from Merck (Darmstadt, Germany). Folin–Ciocalteu reagent, and potassium hexacyanoferrate (III), gallic acid (990 g·L^−1^) were purchased from Sigma (Sternheim, Germany). Sodium carbonate anhydrous was obtained from BDH (Poole, UK), while 2,4,6-tris (2-pyridyl)-*S*-triazine (TPTZ, 990 g·L^−1^) and ferric chloride were acquired from Fluka (Buchs, Germany), and anhydrous sodium sulphate from Panreac (Barcelona, Spain). Ethanol absolute anhydrous was purchased from Carlo Erba (Marseille, France). Luteolinidin chloride was acquired from Chromadex (Los Angeles, CA, USA). Maltodextrin (DE 16.5–19.5) and Arabic gum was purchased from Sigma-Aldrich (Darmstadt, Germany). All other unlabelled chemicals and reagents were analytical, or HPLC-MS Optima grade from Fisher Scientific (Waltham, CA, USA).

### 2.2. Plant Material

Sorghum was purchased dried from the company PMA28 (Varize, France), and ground in a mill (IKA Micro Fine Mill Culatti) using a 1.0 mm thick sieve, and stored in glass bottles (Schott 250 mL) inside desiccators containing silica gel and kept in the dark, at room temperature, until further analysis.

### 2.3. Flavonoids Laboratory-Scale Extraction

The 3-deoxyanthocyanidins dye colourants were extracted according to [24] by mixing the powered plant with 70% ethanol/30% water (containing 1 mL HCl/100 mL), at a ratio of 1:30 (*w*/*v*) material-to-liquor. The solution was placed in an orbital shaker (Unitronic-OR, Selecta, Barcelona) at 25 °C, stirring at 70 rpm, for 2 h. The extracts were placed in a refrigerated centrifuge (Sigma and Laborzentrifugen, 1k15) working at 3100 rpm, 5 °C and for 20 min. The supernatant was separated from the residue and the solvent was evaporated under vacuum (40 °C, 178 mbar) in a rotary evaporator (Buchi R-114 Rotavapor Vap System). Freeze-drying of the extracts was performed in a freeze-dryer (Scanvac Cool Safe, Labogene Scandinavian by Design). The powder was weighed, and placed in glass flasks (Schott 250 mL), at room temperature, under darkness in desiccators. The yield was determined according to Equation (1) [25]. The assays were performed in duplicate.
(1)$$Yield\left(\%\right)=\frac{Mass\;of\;sorghum\;residue}{Mass\;of\;husk\;powder}\times 100$$

### 2.4. Encapsulation of Red Dye by Spray-Drying (SD) and Freeze-Drying (FD)

Different formulations were prepared by using the two carrier agents at different ratios (2:1, 1:1 and 1.5:2.5 *w*/*w*) between of the active agent and the carrier agent. Stock solutions of 40% (*w*/*w*) maltodextrin or 35% (*w*/*w*) Arabic gum were used as carrier agents to produce the water-soluble shell-coated matrix capsules according to the methodology of [17]. Each solution was placed on a hot plate at 60 °C with stirring at 200 rpm until complete dissolution. Then 20% (*w*/*w*) was added to the colourant extract with the pH adjusted to 3.5. The extract solution was placed on a hot plate and heated at 45 °C with stirring at 4500 rpm for one hour. Half of the extracts encapsulated and non-encapsulated were frozen at −80 °C and dried in a laboratory freeze-dryer (Scanvac Cool Safe, Labogene Scandinavian by Design), using the operating conditions referred to in [26].

The other half of the non-encapsulated colourant extract was used in the selected formulation (through the freeze-dried microparticles that had the best ratio between the dye extract and the tested carrier agents characterised in the physicochemical tests) and dehydrated in a laboratory-scale spray-dryer (Buchi B-290, Labortechnik AG, Flawil, Switzerland), according to the operating conditions according to [26].

The microparticle powders obtained by freeze-drying and spray-drying were stored in glass flasks (Schott 250 mL), in a desiccator containing silica gel, under darkness at room temperature.

### 2.5. Dye Quality Evaluation

#### 2.5.1. UV–VIS Spectrometry

The total content of 3-DXA (TCA) in the red dye extracts, encapsulated and non-encapsulated, were calculated based on the absorbance, measured at 480 nm, in a UV–Vis spectrophotometer (double-beam; Hitachi U-2010), between 350 and 550 nm at 1 nm sampling intervals with an average scan rate, estimated according to the Beer–Lambert law and expressed as equivalent mg L^−1^ of luteolinidin, with the following Equation (2) [27]:(2)$$TCA\left(mg\;L^{-1}\right)=\frac{A\times M\times DF\times 10^{3}}{ɛ\times l}$$
where A = Abs; M = 270: the molecular weight of luteolinidin (g mol^−1^); DF: the dilution factor; 10^3^: conversion factor from g to mg; (ε) = 29.157: the molar extinction coefficient of luteolinidin (Lmol^−1^ cm^−1^); l: path length (which is 1 cm).

#### 2.5.2. Liquid Chromatography and Tandem Mass Spectrometry

The extracts were analysed by HPLC-DAD-MS Dionex Ultimate 3000SD on a diode array detector, coupled online to an LCQ Fleet ion trap mass spectrometer, equipped with an ESI source, operating in the positive mode (Thermo ScientificTM, Waltham, MA, USA). Spectra with an average of 20–35 scans were recorded between 100 and 1000 Da. The wavelength was monitored between 250 and 700 nm. Data acquisition and processing were performed using the Xcalibur 2.2 SP1.48 software.

High-resolution tandem mass spectra were obtained on a UHPLC Elute interfaced with a QqTOF Impact II mass spectrometer equipped with an ESI source, operating in the positive mode (Bruker Daltonics, Bremen, Germany). Internal calibration was achieved with an ammonium formate solution introduced to the ion source via a 20 μL loop at the beginning of each analysis, using a six-port valve. The acquisition was performed in a data-dependent MS/MS mode with an acquisition rate of 3 Hz using a dynamic method with a fixed cycle time of 3 s and an *m*/*z* dependent isolation window of 0.03 Da. Data acquisition and processing were performed using Data Analysis 4.2 software.

In both types of equipment, the chromatographic separation was achieved on a Kinetex C18 column 100 Å (150 × 2.1 mm, 2.6 μm particle size, Phenomenex) at 40 °C, using a flow rate of 0.3 mLmin^−1^. The mobile phase was 0.1% of acid formic in water (*v*/*v*, eluent A) and in acetonitrile (eluent B), the elution gradient was as follows: 0–2 min linear gradient to 7% B; 2–23 min linear gradient to 100% B; 23–27 min isocratic 100% B; 27–30 min linear gradient to 7% B; and then the column was re-equilibrated with 7% B for 7 min.

### 2.6. Biological Evaluation of Red Dye from Sorghum

The antioxidant activity of the sorghum extract was performed using two biological in vitro methods: the cell-based OxHLIA (oxidative haemolysis inhibition) assay and TBARS (thiobarbituric acid reactive substances) assay, used for lipid peroxidation inhibition assessment. The synthetic antioxidant Trolox was used as a positive control for both assays, and the procedure was followed as described previously [28]. The extract was dissolved in PBS to obtain the range of concentrations tested (1400–43.75 µg mL^−1^). Results were expressed as the extract concentration responsible for keeping 50% of the erythrocyte population intact (IC_50_ µg/mL) at Δt of 60 min. The ability of the sorghum extract to inhibit TBARS formation was evaluated following the procedure described previously [29]. The extract was dissolved in water to obtain the range of concentrations tested (1400–43.75 µg mL^−1^). The results were expressed as the extract concentration that causes a 50% inhibition of the oxidative process (IC_50_ µg mL^−1^).

The cytotoxicity of the sorghum extract was performed by determining the growth inhibition activity in four tumoral cell lines: AGS (gastric adenocarcinoma), CaCo-2 (colorectal adenocarcinoma), MCF-7 (breast carcinoma) and NCI-H460 (non-small cell lung cancer), and in a non-tumoral primate cell line: VeRO (kidney, African green monkey), using the Sulforhodamine B assay, according to [30]. Briefly, cells were plated in 96-well micro-plates exposed to different concentrations of the extract (400–3.125 µg mL^−1^). The plates were then incubated at 37 °C for 24 h in optimal cell growthconditions. Ellipticine was used as a positive control, and results were expressed as GI_50_ values (concentration required to inhibit 50% of cell growth). The anti-inflammatory potential was assessed using a cell-based model of lipopolysaccharide (LPS)-stimulated RAW 264.7 (murine macrophage-like cell line) cells, as previously described [31]. For this purpose, the concentrations of nitrite oxide (NO) produced by LPS-stimulated RAW 264.7 cells were quantified in different extract concentrations (400–3.125 µg mL^−1^). NO levels were determined by measuring nitrite production spectrophotometrically using the Griess reagent system kit, and the results were expressed as IC_50_ values (extract concentration achieving 50% inhibition of NO production).

### 2.7. Microparticles’ Characterization

#### 2.7.1. Encapsulation Yield (EY), Encapsulation Efficiency (EE), Solubility and Total Phenolic Compounds (TPC)

The encapsulation yields (EYs) of the red extract, in maltodextrin and Arabic gum in the different formulations, were calculated according to the methodology described by [32], using Equation (3):(3)$$EY\left(\%\right)=QE\times 100/ER+E$$
where QE is the 3-DXA content (g), ER the non-encapsulated colourant and E the carrier agent (g).

The encapsulation efficiency (EE) is the relationship between the phenolic compounds’ content (TPC) from a known amount of powder particles after rupture and the phenolic compounds’ content on the surface (SPC) of the same amount of powder particles. To determine the EE, the experimental process was divided into two parts, according to [26]. The encapsulated phenolic compounds were released from microcapsules with a sodium citrate buffer solution at pH 8. The non-encapsulated phenolic compounds were extracted with ethanol. In both cases, the quantification of encapsulated and non-encapsulated total phenolic compounds was determined by the Folin–Ciocalteu method.

The encapsulation efficiency (EE), the solubility, the TPC and the SPC were calculated between the encapsulated and the non-encapsulated red colourant extract, according to the methodology described by [26]. The EE was calculated using Equation (4):(4)$$EE\left(\%\right)=\left(TPC-SPC\right)/TPC\times 100$$

#### 2.7.2. Colourimetry

The 3-DXA extract’s colourimetric evaluation was performed by measuring the parameters colour L* (luminosity between 0—black and 100—white), a* (reddish-green), b* (yellowish-blue) using a colourimeter Chroma meter CR-400 (Konica Minolta, Japan). The powder microcapsules colour measurements were determined with a CS-5 CHROMA SENSOR spectrophotometer (Datacolor International), using an illuminant D 65 and a 2° observation angle. The total colour difference degree (ΔE*) between the extracts of 3-DXA encapsulated and non-encapsulated were calculated according to [33] employing the formula: ΔE* = √ [(ΔL*)^2^ + (Δa*)^2^ + (Δb*)^2^] where ΔL*, Δa* Δb* represent the difference between each parameter for the 3-DXA extracts, where the non-encapsulated red colourant extract was used as control. The measurements were realised in triplicate.

#### 2.7.3. Morphology and Particle Size Distribution

The morphology of the particles was observed by scanning electronic microscopy (SEM). Each sample was covered with a fine layer of gold through the Sputter Coating Attachment of Quorum Q150R ES in vacuumed evaporators. The equipment used for observations was a scanning electron microscope (Hitachi, S-3400N, Tokyo, Japan) with a voltage of 20 kV. The microphotographs were carried out with a camera coupled to the microscope. The samples were systematically observed with 1000× and 3000× magnification. The particle size was determined by examination of SEM micrographs, according to [17].

### 2.8. Stability Tests of the Red Colourant Non-Encapsulated and Encapsulated Released from Microparticles

#### 2.8.1. Effect of pH

A fraction of the microparticles’ powder (0.1 g), maltodextrin and Arabic gum in the different formulations, loaded with the red extract, were diluted with buffer solutions for pH 3, 4, 5 and 6. The buffer solution was prepared by mixing 0.27 g of sodium citrate dehydrate (294.0 g mol^−1^) and 1.74 g of citric acid (192.12 g mol^−1^), and distilled water was added to achieve a volume of 100 mL NaOH (1 M) solution was used to adjust the pH. The colourant-free solution was used as a control. The colourant extracts, encapsulated and non-encapsulated, were diluted to obtain initial absorbance between 0.6 and 0.8 at maximum absorbance length. Then 5 mL of the previous solutions were transferred into glass vials (10 mL) and hermetically sealed. The scanning spectra of the solutions were plotted at 470 nm using a UV–visible spectrophotometer (double-beam; Hitachi U-2010). Citrate buffer solutions were used to constitute the blank solutions.

The samples were used to determine the colour loss of and the highest total concentration of 3-DXA between the encapsulated and the non-encapsulated dye. The assays were performed in duplicate.

#### 2.8.2. Effect of Temperature

Microparticle samples approximately 0.03 g, loaded with red sorghum dye, were weighed into volumetric flasks (25 mL) and completed with a buffer solution at pH 6 (elected in Section 2.8.1). The non-encapsulated colourant was used as control in the temperature stability tests. Then 5 mL of the extracts previously prepared were transferred into 100 mm × 150 mm clear glass vials (10 mL) and hermetically sealed. The vials were placed in a thermostat bath (Unitronic-OR, Selecta, Barcelona) at 80 °C and duplicate vials were removed at 1.5, 3, 6, 9, 12, 15 and 18 h, and immediately cooled in an ice bath to stop thermal degradation. Determination of the of 3-DXA total concentration, encapsulated and non-encapsulated, was performed by measuring the absorbance at 470 nm using a UV–visible spectrophotometer (double-beam; Hitachi U-2010) to evaluate changes in the of 3-DXA total concentration and the difference in colour between the encapsulated and non-encapsulated dye during the heat treatment. The assays were performed in mass duplicate.

#### 2.8.3. Kinetic Parameters

The absorbance measurements of the dye dispersions obtained in Section 2.8.2 were used to obtain the kinetic parameters according to [34] to estimate the stability of the 3-DXA under the action of heat temperature. The first-order reaction rate constants (K_d_), half-lives (t_1/2_), i.e., the time that is necessary for degradation of 50% of 3-DXA, were calculated by the following Equations (5) and (6):(5)$$\ln(C_{t}/C_{0})=-k\times t$$
(6)$$t_{1/2}=-\ln(0.5)\times k^{-1}$$
where C_0_ is the 3-DXA initial concentration, C_t_ the 3-DXA concentration after t minutes of heating at a given temperature.

### 2.9. Statistical Analysis

The results were submitted to one-way analysis of variance (ANOVA) using multiple comparison tests (Tukey HSD) to identify differences between groups. Statistical analyses were tested at a 0.05 level of probability. The range, mean and relative standard deviation (RSD) of each parameter were calculated, and overall variation was assessed by principal component analysis (PCA) of the data obtained from the 10 physicochemical parameters and performed using the software, StatisticaTM 8.0 (Statsoft, Inc., Tulsa, OK, USA, 2007).

## 3. Results

### 3.1. Deoxyanthocyanins’ Profile by Liquid Chromatography–Tandem Mass Spectrometry

The identification of the red chromophores present in the sorghum extract was achieved by HPLC-DAD together with high resolution tandem mass spectrometry analysis. Compounds were identified based on their UV–VIS data, and on the accurate *m*/*z* values of the cationic species. Probable ionic formulas were validated by extracting the ion chromatograms from the raw data, and the accurate mass, isotopic pattern, and fragmentation paths were evaluated. As mentioned in the literature [35,36], the most common anthocyanins in sorghum are 3-DXA and their derivatives, and due to the lack of a hydroxyl group at the C-3 position (Figure 1), they occur mainly in plants in the form of aglycones. In addition, two dimeric 3-DXA were identified in the *Sorghum bicolor* L. extract [37]. The HPLC-DAD chromatogram of the sorghum extract recorded at 470 nm clearly indicates that the most abundant 3-DXA is apigeninidin (Appendix A). The 3-DXA were tentatively identified based on their fragmentation pathways, and by comparison with data from the literature. The results are summarized in Table 1.

### 3.2. In Vitro Antioxidant Activity, Cytotoxicity, Anti-Inflammatory Activity

The extract was assessed for different bioactivities, including antioxidant activity, cytotoxicity (in normal and tumoral cell lines) and anti-inflammatory activity (Table 2). Moderate antioxidant activity for both antioxidant assays was exhibited, with IC_50_ values of 803 μg mL^−1^ (for OxHLIA) and 1210 μg mL^−1^ (for TBARS). Thus, the extract revealed a capacity to inhibit oxidative haemolysis in integral cells and the lipid peroxidation process. These capacities are always valuable to find in natural extracts. Previous studies focusing on different sorghum varieties’ antioxidant capacities only employed chemical methods such as DPPH, ABTS or FRAP, among others [13,38]. Because the methods used in this study were biological and acknowledged as more relevant in assessing natural extracts’ antioxidant activity than chemical methods, a comparison was not considered relevant.

Across all cell lines, there was no observable cytotoxicity, with the highest concentration of sorghum extract used (400 µg mL^−1^) not presenting cell proliferation inhibition. This result indicates that the sorghum extract probably does not affect cell functioning and behaviour and is safe to use at the concentrations tested.

The sorghum extracts’ anti-inflammatory activity was tested by assessing its ability to decrease NO production in LPS-stimulated RAW 264.7 cells. NO production is a good indicator of the inflammatory process as it plays a vital role in the pathogenesis of inflammation. The sorghum extract presented no observable inhibition of NO radical production for the highest concentration analysed (400 µg mL^−1^), indicating that the compounds present in the extract could not inhibit the mechanisms of NO production.

### 3.3. EE, EY, Solubility and TPC

The function of the carrier agent is to protect the active agent, and EE is fundamental to obtaining the microparticles for any encapsulation process. For the freeze-drying process, significant differences (*p* < 0.05) can be observed in the EE of 3-DXA, encapsulated in maltodextrin and Arabic gum for all ratios. The EE of 3-DXA encapsulated in Arabic gum showed higher values than that of 3-DXA encapsulated in maltodextrin (Table 3). However, no significant differences (*p* > 0.05) were observed for the same encapsulating agent at different ratios. For the spray-drying process, no significant differences (*p* > 0.05) were seen between the two encapsulating agents for the 2:1 ratio.

These results agree with those reported by [15] and showed that it could be due to the interaction of hydrogen bonds of the 3-DXA compounds with the glycoproteins of Arabic gum, resulting in stable complexes [15]. They also reported that the different structures of phenolic compounds influence the degree and manner of interaction with polysaccharides. In maltodextrin, the OH groups of the carbohydrate chain interact via hydrogen bridge bonds with the 3-DXA compounds and may lead to less stable complexes and dispersion in the powdered microparticles.

Regarding EY, significant differences were observed (*p* < 0.05) for all samples, with maltodextrin encapsulating agent showing the most retained 3-DXA for all ratios and independently of the drying process, leading to a higher colourant stabilisation during the drying process. These differences can be explained by the concentration of wall material used, as higher concentrations can cover more 3-DXA molecules and improve EY. The authors of [39] found that EY and the properties of the encapsulated agents depended on the retained compounds and the proportion of carrier agents used to encapsulate the prickly pear juice through the spray-drying process.

The solubility results (Table 3) did not show significant differences (*p* > 0.05) among all samples. However, maltodextrin showed the highest solubility for all ratios because the existence of OH groups in their molecular structure, through the interaction of hydrogen bridge bonds with the water molecules, makes them more soluble. Arabic gum has a hydrophobic molecular structure, making this interaction difficult and consequently less soluble.

The results obtained for TPC showed significant differences (*p* < 0.05) in the parameters evaluated for the extracts released from the two carrier agents, for all ratios, and between the two encapsulation processes. The powders with the highest TPC presented the 2:1 ratio with no significant difference (*p* > 0.05) for the samples with different wall materials and encapsulation processes. However, there were no significant differences (*p* > 0.05) for the TPC obtained in the extracts, for the 2:1 ratio, for maltodextrin and Arabic gum, obtained by spray-drying and freeze-drying, respectively. Moreover, no significant differences (*p* > 0.05) were observed for the TPC of the extracts released, in the ratio 1.5:2.5, of maltodextrin and Arabic gum obtained by freeze-drying. However, as reported by [17], the results obtained for the non-encapsulated and encapsulated extracts may not correspond to the actual values since the dispersion of the colouring fraction may not occur during the hydration process, which in turn may lead to the incomplete release of the colouring fraction.

### 3.4. Colour Parameters of Red Dye in Released Tests at Different pH Values

Figure 2 shows, for the different pH values, a significant difference (*p* < 0.05) in the total colour difference (ΔE*) of the encapsulated and non-encapsulated extract obtained for the two encapsulating agents and different ratios obtained by the two drying processes. The results indicated that the lowest ΔE* was obtained for the 2:1 ratio for both carrier agents and drying process, with no significant differences (*p* > 0.05) for pH 6. For the other ratios, the highest ΔE* was observed for the ratios (1.5:2.5) and showed significant differences (*p* < 0.05) for all pH values.

The concentration of the colourant at different ratios of active agent to encapsulant with the variation in pH affects the colourant stabilisation. We found that the higher the pH values and colourant agent ratio to encapsulant, the lesser the difference in the colour total between the encapsulated and non-encapsulated extracts. The formation of complexes between the dye and the polymeric material affects colourant stabilisation positively. Ref. [15] reported that at a less acidic pH, Arabic gum is more effective in apigeninidin stabilisation, the majority compound of the extract, than luteolinidin. The absence of the OH group at C3 in 3-DXA makes the C4-C5 region more hydrophobic when compared to anthocyanins, enhancing the interaction of 3-DXA with the Arabic gum glycoprotein and resulting in better stabilisation.

Furthermore, according to [9], the balance between the flavylium cation and the quinoidal base colour forms results in a colour change from yellow to red–purple, closer to the colour of the non-encapsulated extract.

The various carrier agents to encapsulate beet and saffron extract by freeze-drying showed that maltodextrin and Arabic gum retained colour better, and offering higher protection to other carriers agents after ten weeks of storage were studied by [40].

The results obtained for TCA (Figure 3) determined in the non-encapsulated and encapsulated extract for the two carriers obtained by freeze-drying and spray-drying, for the different ratios and pH values, showed a significant difference (*p* < 0.05). For all ratios, the highest TCA values were obtained in the extracts encapsulated with Arabic gum for the freeze-drying process. These results show the same trend as EE, indicating that the encapsulated compounds were more retained in the Arabic gum than in the maltodextrin, and when released from the microparticles, showed the highest TCA. The highest TCA was found for the 2:1 ratio, followed by the 1:1 ratio, and the lowest TCA was at the 1.5:2.5 ratio for both carriers.

As for the TCA values, determined at pH 4, 5 and 6, there were no significant differences (*p* > 0.05) for the 1.5:2.5 ratio for the freeze-drying process.

At pH 6, TCA showed significant differences (*p* < 0.05) only for the 2:1 ratio of the extracts encapsulated with Arabic gum obtained by freeze-drying.

Regarding the freeze-drying process, no significant differences (*p* > 0.05) were observed between the two carriers and for all pH values.

Similar results were observed by [41] for beetroot extract encapsulated in Arabic gum and maltodextrin, obtained by freeze-drying, where the Arabic gum presented the highest pigment content, due to its low hygroscopicity, compared to the dyes encapsulated with maltodextrin.

### 3.5. Correlations between Physicochemical Parameters

The correlation coefficients were calculated for the ten parameters analysed (Table 4) to evaluate the relationships between the intensity of the association observed and to relate the casual relationships of the variables studied. A positive correlation (*p* < 0.01) was obtained between dye concentration (Dye Conc.) and TPC and negatively with all the variations in the colour parameters of the released compounds analysed at different pHs. Negative correlations between EE and EY, and with the variations in the colour powder parameters (ΔE* colour powder) were observed. EY was positively correlated only with the ΔE* colour powder parameters. Most of the strong correlations detected in the ten samples were confirmed by separating maltodextrin and Arabic gum groups and were mainly due to the different physicochemical parameters studied.

Correlation coefficients were calculated for the ten parameters analysed (Table 4) to evaluate the relationships between the intensity of the observed association and to relate the causal relationships of the variables studied. A positive correlation (*p* < 0.01) was obtained between dye concentration (Dye Conc.) and TPC. A negative correlation was detected for the variations in the colour parameters of the analysed released compounds at different pHs. A negative correlation was observed between EE, EY and the difference in powder colour (E*). EY was positively correlated with the difference in powder colour (E*). The high correlations detected in the ten samples were confirmed by separating maltodextrin and Arabic gum into groups and were due to the different physicochemical parameters studied.

### 3.6. Principal Component Analysis (PCA) to Access Overall Variation

A PCA based on the correlation matrix was performed to allow a better understanding of the interactions between the two encapsulating agents used (maltodextrin and Arabic gum) and the ratio between the compounds to be encapsulated (3-DXA). In the first approach, all ten parameters were included as active variables, and then those factors exhibiting coordinates results of the linear combination of the six active and four supplementary variables, explaining 64.10% and 22.13%, respectively (Figure 4), and corresponding to 86.23% of the total variance. The first principal component was positively correlated with the dye concentration (Dye Conc. (0.95)) and the TPC (0.91); and negatively with a colour variation in the released compounds from microparticles at different pHs, ΔE* pH 3 (−0.93) and ΔE* pH 4 (−0.86), and ΔE* pH 5 (−0.94) and ΔE* pH 6 (−0.93), in line with the previous correlation data. The second component was positively highlighted by the ΔE* powder colour (0.71) and the EY (0.78) indicating that the colour has been retained after processing, and negatively by EE (−0.83). The Dye Conc. and TPC variance were distributed between the two components. Grouping the material type (maltodextrin and the Arabic gum) with the ratio and the active agent (Figure 5) we can observe differences. Based on the first component, the 2:1 ratio, on the right side with higher EY and ΔE* powder colour, parameters are separated from the other carrier agent ratios (1:1 and 1.5:2.5 (*w*/*w*)) located on the left side with Dye Conc. and TPC.

The results indicate that maltodextrin in a 2:1 ratio at pH 6, considering the ten parameters analysed, is the most suitable polymeric material to encapsulate 3-DXA, as it can give rise to powder dyes with a colour closer to the non-encapsulated colourant. Moreover, in the spray-drying process, maltodextrin 2:1 presents higher resistance to thermal degradation and therefore is more suitable for scale-up.

However, Arabic gum has a higher EE due to its affinity for 3-DXA dye structures. In addition, adjusting the spray-drying process conditions could increase the EY and heat resistance, increasing the application of these powder colourants.

### 3.7. Profile of Encapsulated Red Dye Extracts in Released Tests

Figure 6 illustrates the variation in the relative peak areas of each red dye present in the freeze-dried sorghum microcapsules as a function of encapsulation with maltodextrin and Arabic gum obtained by freeze-drying and spray-drying processes.

The results indicate that we cannot observe significant differences in the retention of 3-DXA by the two carrier agents. Relative to the processes used to obtain the microparticles, it was the freeze-drying process that allowed for better retention of the 3-DXA compounds.

### 3.8. Morphology and Particle Size Distribution by Scanning Electron Microscopy (SEM)

The microphotographs presented in Figure 7 show the morphologies of the microparticles obtained for the microencapsulation of the 3-DXA produced with maltodextrin (A) and Arabic gum (B) (in a 2:1 ratio) by spray-drying. The sample’s surface shows rounded microparticles with a continuous wall and some wrinkles in the structures but no cracks. The authors of [42] observed similar morphologies in microparticles of anthocyanidins obtained with different types of polysaccharides by spray-drying. The particles size distribution was calculated according to [17], and ranged from 5 mm to 40 mm approximately, for both carrier agents.

As for the microparticles wall, it was found that they presented a continuous structure, indicating that the encapsulation process was perfect, showing the physicochemical interaction that occurs between the polysaccharides and the 3-DXA pigments. On the other hand, wrinkling in the wall structure indicates that the microencapsulation was not as perfect due, for example, to lower molecular affinity, as found in [42] with different types of polysaccharides.

The freeze-dried particles showed an amorphous structure, figures not shown, possibly due to the rapid evaporation, and the freezing temperature that occurred during the process, which impeded the ability to form crystalline structures, as was reported by [26].

### 3.9. Colour Parameters and Degradation Kinetics of Red Dye in Released Tests

The results obtained for the kinetic degradation, of the non-encapsulated and encapsulated red dye extracts (elected in Section 3.6), corresponding to the maximum absorbance values, as a function of temperature, are represented by the linear regression equation and the degradation constants (K_d_) and half-life time (T_1/2_) presented in Table 5. These results show that the longer the temperature exposure time, the higher the degradation constant and, consequently, the shorter the half-life time, indicating further degradation of the 3-DXA compounds. Thus, for pH 6, at 80 °C and 18 h of temperature exposure time, the non-encapsulated extracts showed the highest values for the degradation constants (K_d_ = 0.04), and the lowest for the half-life time (T_1/2_ = 19.94), indicating that the non-encapsulated extracts were more degraded than those obtained with encapsulation.

For the non-encapsulated red dye extracts obtained by freeze-drying, there was a large decrease (about 48%) in the total concentration of 3-DXA (Figure 8), while for the extracts encapsulated by the freeze-drying process, the percentage decrease in anthocyanidin compounds was lower, about 22%, for both encapsulating agents. The same decrease trend, about 28%, was observed for the spray-drying process for both encapsulating agents.

The results obtained for the kinetic degradation of the non-encapsulated and encapsulated red dye extracts (elected in Section 3.6), corresponding to the maximum absorbance values, as a function, of temperature, are represented by the linear regression equation and the degradation constants (K_d_) and half-life time (T_1/2_) presented in Table 5. These results show that the longer the temperature exposure time, the higher the degradation constant and the shorter the half-life time, indicating further degradation of the 3-DXA compounds. Thus, for pH 6, at 80 °C and 18 h of temperature exposure time, the non-encapsulated extracts showed the highest values for the degradation constants (K_d_ = 0.04), and the lowest for the half-life time (T_1/2_ = 19.94), indicating that the non-encapsulated extracts were more degraded than those obtained with encapsulation.

For the non-encapsulated red dye extracts obtained by freeze-drying, there was a higher decrease in the total concentration (about 48%) of 3-DXA (Figure 8). For the extracts encapsulated by the freeze-drying process, there was a smaller percentage decrease in anthocyanidin compounds, about 22%, for both encapsulating agents. The same decrease trend, about 28%, was observed for the spray-drying process for both encapsulating agents.

The encapsulation process has a positive effect on the dye thermal stabilization due to the formation of stable complexes between 3-DXA and polymeric materials. Other studies [15,43] had already observed that solutions of the 3-DXA dye were unstable in an aqueous medium at pH 5, precipitating the dye, and the addition of Arabic gum improved the dye stabilization.

## 4. Conclusions

The *Sorghum bicolor* L. extract showed moderate antioxidant activity, with EC_50_ values between 803 and 1210 μg L^−1^ (for OxHLIA and TBARS assays, respectively), which encourages its use as a dye with health benefits, and did not show anti-inflammatory or cytotoxic properties at the maximum concentration studied (400 μg mL^−1^), thus being considered safe for food application. It was shown that the variability in the ratio composition of the two carrier agents used to encapsulate the 3-DXA could be discriminated by the type of polymer used and the colour variation in the compounds released from the microparticles at different pH values. In addition, the powder colour and the yield of the encapsulation process contributed to the selection of the best carrier. A higher content of dye and TPC was found in 2:1 maltodextrin at pH 6.

The microencapsulated extracts analysed by LC-DAD-ESI-QTOF/MS showed a high diversity of 3-DXA, especially apigeninidin, the majority compound, when encapsulated using the freeze-drying process. Regarding the stability of 3-DXA at 80 °C, during 18 h, it was found that the dye extracts encapsulated with maltodextrin and obtained by freeze-drying maintained the structure and were less degraded (22%) than those obtained by spray-drying. The more regular microparticles’ shape in the spray-drying process may improve the stability of the red dye extracts and allow for larger-scale production in the food industry.

Obtaining the red colourants from the leaves of the sorghum culture may allow the use of non-edible parts of the plant, increasing the valorisation of this crop. On the other hand, the process of the encapsulation of red dye extracts allows for the degradation of the dye to be stabilised at neutral pH and increases the scale of application as an ingredient for the food industry.

## Figures and Tables

**Figure 1 foods-12-01646-f001:**
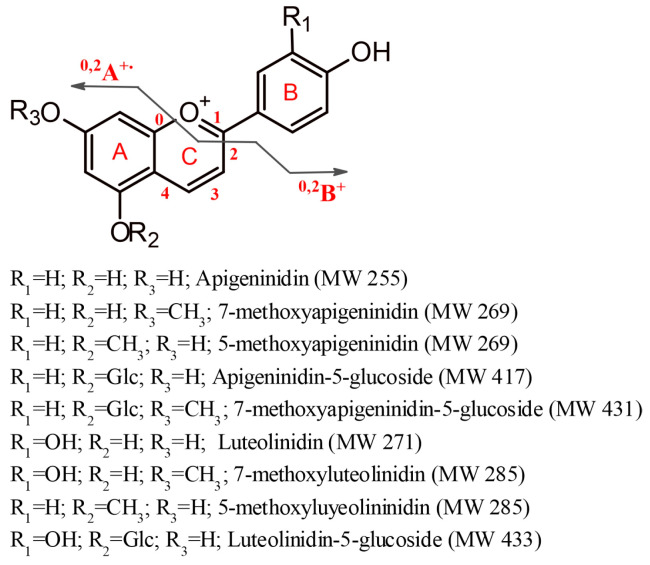
Chemical structures and molecular weights of 3-DXA and derivatives present in red sorghum. Nomenclature and two characteristics fragment ions of anthocyanins are also illustrated in the figure.

**Figure 2 foods-12-01646-f002:**
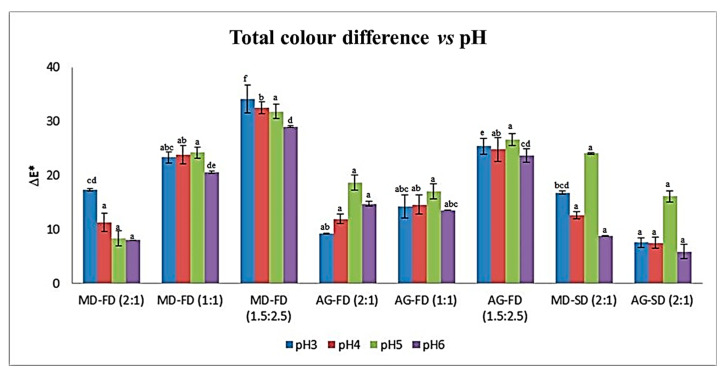
Total colour difference (ΔE*) for the red dye extract non-encapsulated (NE) and extracts encapsulated with maltodextrin (MD) and Arabic gum (AG) obtained by freeze-drying (FD) and by spray-drying (SD) (mean ± SD). Different letters indicate significant differences (*p* < 0.05) within the same ratio for carrier agents and the encapsulation process.

**Figure 3 foods-12-01646-f003:**
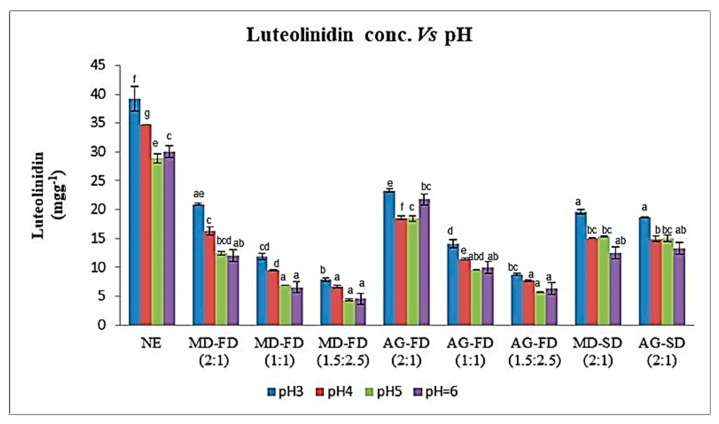
The concentration of luteolinidin (mg g^−1^) for the red dye extract non-encapsulated (NE) and extracts encapsulated with maltodextrin (MD) and Arabic gum (AG) obtained by freeze-drying (FD) and by spray-drying (SD) (mean ± SD), and dispersed at different pH values (3–6). Bars with the same letters are not significantly different at 5%.

**Figure 4 foods-12-01646-f004:**
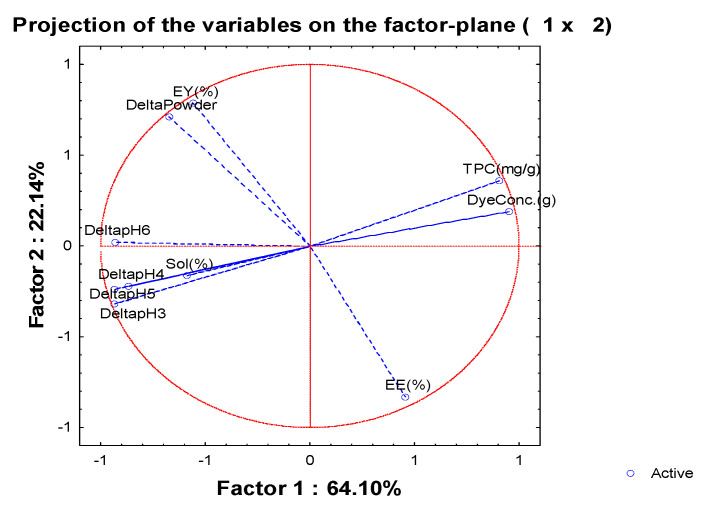
Principal component analysis with six active and four supplementary variables. Abbreviations: encapsulation efficiency (EE), encapsulation yield (EY), dye concentration (Dye Conc.), solubility (Sol), powder colour variation (Δ powder), colour variation at pH 3, pH 4, pH 5 and pH 6 (ΔE* pH 3, ΔE* pH 4, ΔE* pH 5 and ΔE* pH 6).

**Figure 5 foods-12-01646-f005:**
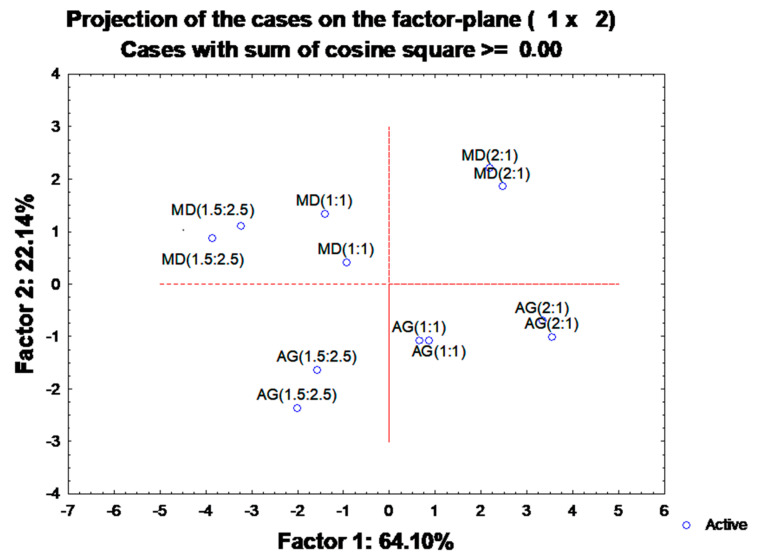
Principal component analysis with the grouping of accessions, according to maltodextrin (MD) or Arabic gum (AG) ratio (2:1, 1:1 and 1.5:2.5).

**Figure 6 foods-12-01646-f006:**
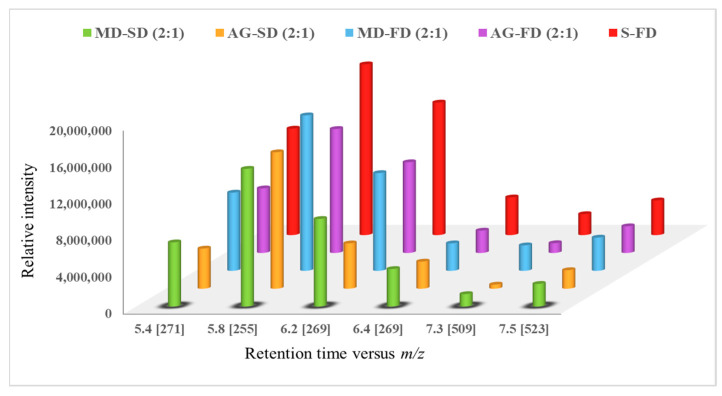
Comparative diagram of the relative abundances of 3-DXA present in the red dye extract non-encapsulated extracts (S-FD) and extracts encapsulated with maltodextrin (MD) and Arabic gum (AG) obtained by freeze-drying (FD) and by spray-drying (SD).

**Figure 7 foods-12-01646-f007:**
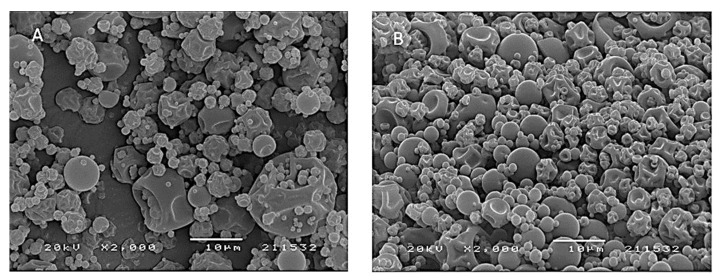
SEM images of sorghum microparticles obtained with spray-drying by maltodextrin (**A**) and Arabic gum (**B**), (magnification 2000×).

**Figure 8 foods-12-01646-f008:**
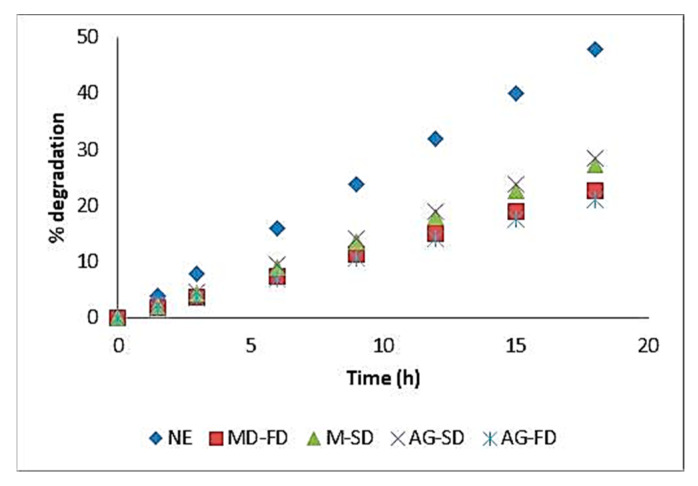
Degradation (%) kinetics in function of time (hours) for the red dye extract non-encapsulated (NE) and extracts encapsulated with maltodextrin (MD) and Arabic gum (AG) obtained by freeze-drying (FD) and by spray-drying (SD) (mean ± SD), at 80 °C.

**Table 1 foods-12-01646-t001:** HPLC-DAD and LC-ESI(+)-HRMS/MS characterisation of the main red chromophores present in the *Sorghum bicolor* L. extract.

Rt (min)	λ Max (nm)	Ionic Formula	[M]^+^ (*m*/*z* (Δ ppm))	MS/MS [(*m*/*z* (Δ ppm) (Attribution)]	Proposed Compound
4.6	-	C_21_H_21_O_9_^+^	417.1190 (−2.4)	[255.0659 (−3.0) [Y_0_^+^] (C_15_H_11_O_4_)^+^] [227.0696 (+3.0) (C_14_H_11_O_3_)^+^]	Apigeninidin-5-Glc
4.7	-	C_22_H_23_O_9_^+^	431.1347 (−2.4)	[269.0816 (−2.9) [Y_0_^+^] (C_16_H_13_O_4_)^+^] [254.0578 (−1.6) (C_15_H_10_O_4_)^+●^] [226.0630 (−2.4) (C_14_H_10_O_3_)^+●^]	7-*O*-methoxy-apigeninidin-5-Glc
5.4	484	C_15_H_11_O_5_^+^	271.0601(−0.1)	[241.0500 (−1.9) (C_14_H_9_O_4_)^+^] [168.0569 (−0.4) (C_12_H_8_O)^+●^] [150.0316 (−1.4) (C_8_H_6_O_3_)^+●^] [137.0226 (+5.3) [^0,2^B^+^] (C_7_H_5_O_3_)^+^] [134.0352 (+8.0) [^0,2^A^+●^] (C_8_H_6_O_2_)^+●^]	Luteolinidin
5.8	472	C_15_H_11_O_4_^+^	255.0652 (−1.2)	[227.0692 (+4.7) (C_14_H_11_O_3_)^+^] [213.0537 (+4.5) (C_13_H_9_O_3_)^+^] [197.0597 (+0.1) (C_13_H_9_O_2_)^+^] [171.0438 (+1.5) (C_11_H_7_O_2_)^+^] [157.0647 (+0.5) (C_11_H_9_O)^+^] [134.0355(+5.1) [^0,2^A^+●^] (C_8_H_6_O_2_)^+●^] [121.0287 (+7.8) [^0,2^B^+^] (C_7_H_5_O_2_)^+^]	Apigeninidin
6.2	472	C_16_H_13_O_4_^+^	269.0812 (−1.5)	[254.0574 (−0.3) (C_15_H_10_O_4_)^+●^] [226.0621 (+1.7) (C_14_H_10_O_3_)^+●^] [197.0600 (−1.6) (C_13_H_9_O_2_)^+^] [169.0647 (+0.8) (C_12_H_9_O)^+^] [144.0566 (+2.7) (C_10_H_8_O)^+●^] [121.0277 (+5.6) [^0,2^B^+^] (C_7_H_5_O_2_)^+^] [115.0539 (+3.2) (C_9_H_7_)^+^]	7-*O*-methoxy-apigeninidin
6.4	474	C_16_H_13_O_4_^+^	269.0814 (−2.0)	[254.0577 (−1.4) (C_15_H_10_O_4_)^+●^] [226.0617 (+3.1) (C_14_H_10_O_3_)^+●^] [197.0602 (−2.3) (C_13_H_9_O_2_)^+●^] [169.0649 (−0.4) (C_12_H_9_O)^+^] [141.0697 (+1.4) (C_11_H_9_)^+^] [121.0283 (+0.9) [^0,2^B^+^] (C_7_H_5_O_2_)^+^] [115.0540 (+1.8) (C_9_H_7_)^+^]	5-*O*-methoxy-apigeninidin
6.6	-	C_21_H_21_O_10_^+^	433.1127 (+0–6)	[271.0609 (−2.9) (C_15_H_11_O_5_)^+^] [243.0654 (−0.7) (C_15_H_11_O_4_)^+^]	Luteolinidin-5-Glc
7.3	476	C_30_H_21_O_8_^+^	509.1238 (−1.3)	[355.0979 (−4.1) (C_23_H_15_O_4_)^+^] [255.0654 (−0.9) (C_15_H_11_O_4_)^+^]	Apigeninidin-flavene dimer
7.5	478	C_31_H_23_O_8_^+^	523.1392 (−0.9)	[383.0920 (−1.7) (C_24_H_15_O_5_)^+^] [269.0805 (+1.1) (C_16_H_13_O_4_)^+^] [255.0653 (−0.3) (C_15_H_11_O_4_)^+^]	Apigeninidin-7-*O*-methoxyflavene dimer

Abbreviation: Glc, glucoside.

**Table 2 foods-12-01646-t002:** Antioxidant activity (IC_50_; µg mL^−1^), cytotoxic and anti-inflammatory activity (GI_50_, μg mL^−1^) of sorghum dye.

Test		Sorghum	Trolox *	Ellipticine *
Antioxidant activity	OxHLIA, Δt = 60 min	803 ± 40	21.8 ± 0.2	
	TBARS	1210 ± 62	139 ± 5	
Cytotoxic activity	AGS	>400	−	1.23 ± 0.03
	CaCo2	>400	−	1.21 ± 0.02
	MCF-7	>400	−	1.02 ± 0.02
	NCI-H460	>400	−	1.01 ± 0.01
	VERO	>400	−	1.41 ± 0.06
Anti-inflammatory activity	RAW 246.7	>400	−	6.3 ± 0.4

* Positive control.

**Table 3 foods-12-01646-t003:** Physicochemical parameters (mean ± SD) for the red dye extract non-encapsulated (NE) and extracts encapsulated with maltodextrin (MD) and Arabic gum (AG) obtained by freeze-drying (FD) and by spray-drying (SD) (mean ± SD).

	Ratio	EE (%)	EY (%)	Solubility (%)	TPC (mg g^−1^)
NE-FD	−	−	15.5 ^a^± 0.14	42.77 ^a^ ± 0.00	3.57 ^f^ ± 0.16
MD-FD	2:1	69.93 ^ab^ ± 2.69	73.77 ^h^ ± 0.25	47.37 ^a^ ± 0.02	7.14 ^b^ ± 0.32
MD-FD	1:1	69.59 ^ab^ ± 9.42	72.33 ^g^ ± 0.47	47.78 ^a^ ± 0.01	4.12 ^d^ ± 0.51
MD-FD	1.5:2.5	64.63 ^ab^ ± 1.66	76.39 ^i^ ± 0.43	48.33 ^a^ ± 0.00	2.53 ^c^ ± 0.18
AG-FD	2:1	81.70 ^a^ ± 4.59	63.72 ^d^ ± 0.06	44.60 ^a^ ± 0.05	6.11 ^ab^ ± 0.71
AG-FD	1:1	82.96 ^a^ ± 0.37	68.72 ^f^ ± 0.19	46.35 ^a^ ± 0.00	4.41 ^de^ ± 0.28
AG-FD	1.5:2.5	80.81 ^a^ ± 2.66	67.33 ^e^ ± 0.25	50.81 ^a^ ± 0.01	2.71 ^c^ ± 0.01
MD-SD	2:1	56.42 ^b^ ± 0.07	51.22 ^c^ ± 0.31	46.90 ^a^ ± 0.30	6.73 ^ab^ ± 0.01
AG-SD	2:1	60.66 ^b^ ± 8.92	39.11 ^b^ ± 0.13	46.84 ^a^ ± 1.50	5.59 ^ae^ ± 0.18

Mean ± standard deviation of the analyses in duplicate. Values with different letters per row indicate significant differences (*p* < 0.05).

**Table 4 foods-12-01646-t004:** Correlation coefficients between the data for the ten physicochemical parameters obtained for the microparticles produced by the freeze-drying process * *p* < 0.05; ** *p* < 0.01.

	Dye Conc.(g)	Solubility (%)	EE (%)	TPC (mg g^−1^)	EY (%)	ΔE* pH3	ΔE* pH4	ΔE*pH5	ΔE* H6	ΔE Colour Powder
**Dye Conc. (g)**	1.00									
**Solubility (%)**	−0.56	1.00								
**EE (%)**	0.24	-0.11	1.0							
**TPC (mg g^−1^)**	0.93 **	-0.48	0.16	1.00						
**EY (%)**	−0.42	0.25	−0.84 **	−0.20	1.00					
**ΔE* pH3**	−0.82 **	0.56	−0.58	−0.74 **	0.69 *	1.00				
**ΔE* pH4**	−0.78 **	0.39	−0.54	−0.84 **	0.46	0.82	1.00			
**ΔE* pH5**	−0.62 *	0.27	−0.30	−0.79 **	0.14	0.59	0.92	1.00		
**ΔE* pH6**	−0.78 **	0.36	−0.35	−0.90 **	0.26	0.79	0.94	0.91	1.00	
**ΔE* colour powder**	−0.50	0.27	−0.87 **	−0.35	0.93 **	0.79	0.61	0.33	0.48	1.00

**Table 5 foods-12-01646-t005:** Kinetic values for the red dye extract non-encapsulated (NE) and extracts encapsulated with maltodextrin (MD) and Arabic gum (AG) obtained by freeze-drying (FD) and by spray-drying (SD) (mean ± SD). Linear regression equation, K values, regression coefficient (r^2^), degradation rates values (K_d_) and half-life time (t_1/2_) (mean ± SD), at 80 °C.

	Linear Regression Equation	K_d_	T_1/2_ (Hours)
NE	y = −0.04 x − 0.08 (R^2^ = 0.87)	0.04 ± 0.00	19.94 ± 1.41
MD-FD	y = −0.01 x − 0.20 (R^2^ = 0.87)	0.01 ± 0.00	48.96 ± 0.25
AG-FD	y = −0.02 x − 0.01 (R^2^ = 0,99)	0.02 ± 0.00	42.39 ± 0.18
MD-SD	y = −0.02 x − 0.15 (R^2^ = 0.92)	0.02 ± 0.00	39.84 ± 0.00
AG-SD	y = −0.02 x − 0.05 (R^2^ = 0.88)	0.02 ± 0.00	32.66 ± 1.63

## Data Availability

Data will be made available upon reasonable request to the corresponding authors.

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
