# Peer review of "Encapsulation of Sorghum Leaf Red Dye: Biological and Physicochemical Properties and Effect on Stability"

_foods, 2023, doi:10.3390/foods12081646_

Round 1
Reviewer 1 Report
The Sorghum bicolour L. extract was explored and found that the extracts presented antioxidant activity without anti-inflammatory nor cytotoxic properties, indicating their potential application in food. The encapsulation of extract was also performed with two carrier agents (maltodextrin and Arabic gum) in different ratios. Results showed Sorghum leaf by-products as a red dye could be a promising ingredient for the food industry. My questions and suggestions are as follows:
1. The introduction section should be reorganized focusing on the development of Sorghum bicolour L. extract including the origin, chemical components, bioactivities, as well as applications.
2. The extracts were analyzed by HPLC-DAD-MS and 9 compounds of anthocyanins were identified. I suggest the anthocyanins in the extracts should also be analyzed quantitatively in order to give sound scientific supports for further study or application.
3. The antioxidant activity of the non-encapsulated sorghum extract was assayed, how about the encapsulated extract? Please add it.
4. The experimental data are sufficient, but the discussion is just simple description. Please give more meaningful explanations.
5. The pictures in the manuscript are not very clear. The resolution of the pictures should be improved. There are some overlap in Figure 4and Figure 5.
6. There are minor spelling errors, for example , L429 non-non-encapsulated, abbreviation in Table 3. Please check and correct carefully.
Author Response
Revisor #1
1 - The introduction section should be reorganized focusing on the development of Sorghum bicolour L. extract including the origin, chemical components, bioactivities, as well as applications.
Answer.
The authors tried to incorporate all the suggestions proposed by the reviewer in the introduction, but at the same time, try keeping the journal aim.
2 - The extracts were analysed by HPLC-DAD-MS and 9 compounds of anthocyanins were identified. I suggest the anthocyanins in the extracts should also be analysed quantitatively in order to give sound scientific supports for further study or application.
Answer: the authors agree that the quantitation of the 3-DXAs present in the sorghum extracts is a useful result which can enhance the quality of the research work. But, the difficulty in acquiring an analytical standard in a short time made the introduction of an analytical quantification unavailable. However, a qualitative analysis comparing the variation in the peak areas of each 3-DXAs cationic ion presents in non-encapsulated and encapsulated sorghum extracts was presented. These results indicated that the microencapsulation process protects the core material against environmental conditions improving its stability as well as solubility and bioavailability.
Considering that the qualitative analysis was based on mass spectrometry data, which are not directly related to the concentration of each chromophore present in the extracts, an LC-DAD-MS spectrum acquired in the ESI negative mode for the non-encapsulated sorghum extract was introduced as Supplementary Material. The DAD chromatogram acquired at 470 nm demonstrates that the mostly abundant 3-DXA present on Sorghum bicolor L. is apigeninidin.
3 – The antioxidant activity of the non-encapsulated sorghum extract was assayed, how about the encapsulant extract? Please add it.
Answer: The authors agree that the antioxidant activity of the encapsulated sorghum extract should be performed, and the results could improve the quality of the research work. The antioxidant activity was a property to use as an extra in food applications involving lipid oxidation, such as meat products. However, the extract exhibits moderate antioxidant activity and the determination, in a significant number, of encapsulated samples and their duplicates using cell-based methods would make the process too expensive to perform all the analyses.
4 – The experimental data are sufficient, but the discussion is just simple description. Please give more meaningful explanations.
Answer: The authors improved the discussion in the document.
5 - The pictures in the manuscript are not very clear. The resolution of the pictures should be improved. There are some overlap in Figure 4and Figure 5.
Answer: The resolution of the photographs in the manuscript have been improved but, as they were taken from a statistical programme, the quality is not the best.
6 - There are minor spelling errors, for example , L429 nonnon-encapsulated, abbreviation in Table 3. Please check and correct carefully.
Answer: The words have been corrected as requested.

Reviewer 2 Report
Overall, this is a nice piece of research that describes the potential of maltodextrin and Arabic gum to encapsulate sorghum leaf red dye with high biological activity. My overall appreciation is positive and it falls within the scope of the journal. However, the discussion of results should be improved, thus, a major revision is needed.
ü The quality of the English writing must be upgraded to make it publishable. I have mentioned some grammatical errors in the following comments.
ü Avoid too-long sentences. You can divide them into several short sentences in many parts of your manuscript.
ü The abstract should be a total of about 200 words maximum. Your abstract needs to focus on only the most critical details.
ü Line 27: at different ratios
ü Line 27: This sentence is not clear enough. Please rewrite it to clearly describe that these ratios are related to maltodextrin and Arabic gum or carriers and encapsulant
ü Line 34: total phenolic compounds (TPC)
ü Lines 58-60: Change to “Africa, North America, and Asia being the main producing regions and Nigeria, the United States, Mexico, India, and China are the five largest producers, corresponding to about 70% of the world production.”
ü Line 60: ease of production
ü Line 69: animal feed as well as for the production of bioethanol.
ü Line 79: … anthocyanins that present human health benefits
ü Line 89: Comma before “they are replacing” and start the next sentence with: Therefore, …
ü Line 91-93: Why do you use a huge amount of commas in the sentences in the whole manuscript??
ü Line 104: quite unique behaviour of Arabic gum is …
ü Line 105:… its stabilizing mechanism is comparable
ü Line 111: by the release of these compounds with pH variations and temperature.
ü Line 118: at a reduced cost
ü Line 138: powdered in a mill
ü Line 146: mL
ü Line 146: What is the meaning of (m/v)?? Do you mean W/V??
ü Line 161: of the active agent and the encapsulant. It seems that “were tested” can be deleted in this sentence.
ü Line 259: What does mean extraction residue?
ü Section 2-7-1: How do you measure the SPC?
ü What was the dye residue that is used as a control? Please explain it clearly.
ü Line 305: 100 mm × 150 mm.
ü Line 393: by assessing its ability
ü Line 413: at different ratios.
ü Line 429: non-encapsulated
ü Section 3.3: The discussion is weak. Please explain more regarding the reasons for significant or insignificant results between different formulations.
ü In all figure and table captions: MD-FD and MD-SD
ü Section 3.4: Can the authors explain the effect of pH values and the ratio of carrier to encapsulant on color parameters?
ü Line 532: The authors reported that maltodextrin in the ratio of 2:1 is a polymeric material more suitable to encapsulate the 3-DXAs than Arabic gum, due to the higher molecular affinity of the structural form with the dye. However, Table 3 showed higher EE values for Arabic gum at all ratios. Can the authors explain these contradictory results?
ü Please improve your discussion regarding findings.
ü The quality of the figures should be modified. Please use a larger font and different colors in figure 6.
ü Line 584: (Figure 8)
ü Line 590-593: The space between °C and numbers should e considered.
ü The format of references should follow a similar pattern.
Author Response
Revisor #2
All suggestions proposed along the lines of the document by the reviewer were accepted and modified.
ü The quality of the English writing must be upgraded to make it publishable. I have mentioned some grammatical errors in the following comments.
Answer:
English has been correct.
ü Avoid too-long sentences. You can divide them into several short sentences in many parts of your manuscript.
Answer:
The corrections were done as indicated.
ü The abstract should be a total of about 200 words maximum. Your abstract needs to focus on only the most critical details.
Answer:
The summary is about 250 words, and we think it focuses on the main critical details.
Ü Line 27: at different ratios.
Answer:
The correction has been as proposed.
ü Line 27: This sentence is not clear enough. Please rewrite it to clearly describe that these ratios are related to maltodextrin and Arabic gum or carriers and encapsulant.
Answer:
We think the sentence is clear.
ü Line 34: total phenolic compounds (TPC)
Answer:
The correction has been as proposed.
ü Lines 58-60: Change to “Africa, North America, and Asia being the main producing regions and Nigeria, the United States, Mexico, India, and China are the five largest producers, corresponding to about 70% of the world production.”
Answer:
The sentence was summarised and rewritten
ü Line 60: ease of production.
Answer:
The sentence was summarised and rewritten.
ü Line 69: animal feed as well as for the production of bioethanol.
Answer:
The sentence was summarised and rewritten.
ü Line 79: … anthocyanins that present human health benefits
Answer:
The correction has been as proposed.
ü Line 89: Comma before “they are replacing” and start the next sentence with: Therefore, …
Answer:
The sentence was summarised and rewritten.
ü Line 91-93: Why do you use a huge amount of commas in the sentences in the whole manuscript??
Answer:
The correction has been as proposed.
ü Line 104: quite unique behaviour of Arabic gum is …
Answer:
The sentence was summarised and rewritten.
ü Line 105: its stabilizing mechanism is comparable
Answer:
The sentence was summarised and rewritten.
ü Line 111: by the release of these compounds with pH variations and temperature.
Answer:
The sentence was summarised and rewritten.
ü Line 118: at a reduced cost.
Answer:
The correction has been as proposed.
ü Line 138: powdered in a mill
Answer:
The correction has been as proposed.
ü Line 146: mL
Answer:
The unit wasn't correct by mistake.
ü Line 146: What is the meaning of (m/v)?? Do you mean W/V??
Answer:
The correction has been as proposed.
ü Line 161: of the active agent and the encapsulant. It seems that “were tested” can be deleted in this sentence.
Answer:
The words weren’t removed by mistake.
ü Line 259: What does mean extraction residue?
Answer:
It means after evaporation of the solvent.
The sentence was rewritten.
ü Section 2-7-1: How do you measure the SPC?
Answer:
SPC is the content of phenolic compounds on the surface of the same amount of powder particles and is determined using the Folin Ciocalteau method described by [32].The sentence was rewritten.
ü What was the dye residue that is used as a control? Please explain it clearly.
Answer:
It means the non-encapsulated red colorant extract.
The sentence was rewritten.
ü Line 305: 100 mm ×150 mm.
Answer:
The correction has been as proposed.
ü Line 393: by assessing its ability
Answer:
The correction has been as proposed.
ü Line 413: at different ratios.
Answer:
The correction has been as proposed.
ü Line 429: non-encapsulated
Answer:
The correction has been as proposed.
ü Section 3.3: The discussion is weak. Please explain more regarding the reasons for significant or insignificant results between different formulations.
Answer:
All suggestions proposed for the discussion of the results were accepted and modified.
ü In all figure and table captions: MD-FD and MD-SD
Answer:
All suggestions proposed to figure and table-captions were accepted and modified.
ü Section 3.4: Can the authors explain the effect of pH values and the ratio of carrier to encapsulant on color parameters?
Answer:
The concentration of the colourant at different ratios of active agent to encapsulant with the variation of pH affects the colourant stabilisation in aqueous solution. We found that the higher the pH values and colourant agent ratio to encapsulant, the lesser the difference in the colour total between the encapsulated and non-encapsulated extracts. The formation of complexes between the dye and the polymeric material affects colourant stabilisation positively. [15] reported that at a less acidic pH, Arabic gum is more effective in apigenunidin stabilisation, the majority compound of the extract, than luteolinidin. The absence of the OH group at C3 in 3-DXA makes the C4-C5 region more hydrophobic when compared to anthocyanins, enhancing the interaction of 3-DXA with the AG glycoprotein and resulting in better stabilisation.
Furthermore, according to [9], the balance between the flavylium cation and the quinoidal base colour forms results in a colour change from yellow to red-purple, closer to the colour of the non-encapsulated extract.
ü Line 532: The authors reported that maltodextrin in the ratio of 2:1 is a polymeric material more suitable to encapsulate the 3-DXAs than Arabic gum, due to the higher molecular affinity of the structural form with the dye. However, Table 3 showed higher EE values for Arabic gum at all ratios. Can the authors explain these contradictory results?
Answer:
The results indicate that maltodextrin in a 2:1 ratio at pH 6, considering the ten parameters analysed, is the most suitable polymeric material to encapsulate 3-DXA, as it can give rise to powder dyes with a colour closer to the non-encapsulated colourant. Also, in spray-drying process maltodextrin 2:1 presents higher resistance to thermal degradation therefore is more suitable for scale-up.
However, Arabic gum has a higher EE due to its affinity for 3-DXA dye structures. In addition, adjusting the spray drying process conditions could increase the EY and heat resistance, increasing the application of these powder colourants.
ü Please improve your discussion regarding findings.
Answer:
All suggestions proposed for the discussion were accepted and modified.
ü The quality of the figures should be modified. Please use a larger font and different colors in figure 6.
Answer:
The correction has been as proposed.
ü Line 584: (Figure 8)
Answer:
The correction has been as proposed.
ü Line 590-593: The space between °C and numbers should e considered.
Answer:
The correction has been as proposed.
ü The format of references should follow a similar pattern.
Answer:
The correction has been as proposed.

Round 2
Reviewer 2 Report
The authors have corrected and answered all comments. Minor revision is now needed according to the following comments.
Line 138: powered or powdered in a mill??
Line 146: mL HCl
Section 2-7-1: Please clearly explain the method of separating SPC to measure it by the Folin Ciocalteau method.
In all figure and table captions: spray-drying (SD)
Author Response
Revisor #2 – second round
Line 138: powered or powdered in a mill??
Answer:
The word was replaced by grinding in a mill.
Line 146: mL HCL
Answer:
The unit was corrected
Section 2-7-1: Please clearly explain the method of separating SPC to measure it by the Folin Ciocalteau method.
Answer:
The encapsulation efficiency (EE) is the relationship between the phenolic compounds content (TPC) from a known amount of powder particles after rupture and the phenolic compounds content on the surface (SPC) of the same amount of powder particles. To determine the EE, the experimental process was divided into two parts, according to [32]. The encapsulated phenolic compounds were released from microcapsules with a sodium citrate buffer solution at pH 8. The non-encapsulated phenolic compounds were extracted with ethanol. In both cases, the quantification of encapsulated and non-encapsulated total phenolic compounds was determined by the Folin-Ciocalteau method.
In all figure and table captions: spray-drying (SD).
In all figure and table captions the word SD was corrected.
